# Phylogeography and Evolutionary Dynamics of Tobacco Curly Shoot Virus

**DOI:** 10.3390/v16121850

**Published:** 2024-11-28

**Authors:** Xingxiu Long, Shiwei Zhang, Jianguo Shen, Zhenguo Du, Fangluan Gao

**Affiliations:** 1Institute of Plant Virology, Fujian Agriculture and Forestry University, Fuzhou 350002, China; longxingxiu@fafu.edu.cn (X.L.); swayzhang@fafu.edu.cn (S.Z.); duzhenguo1228@163.com (Z.D.); 2Fujian Key Laboratory for Technology Research of Inspection and Quarantine, Technology Center of Fuzhou Customs District, Fuzhou 350001, China

**Keywords:** tobacco curly shoot virus, Begomovirus, Bayesian phylodynamics, substitution rate, temporal signal

## Abstract

Tobacco curly shoot virus (TbCSV), a begomovirus, causes significant economic losses in tobacco and tomato crops across East, Southeast, and South Asia. Despite its agricultural importance, the evolutionary dynamics and emergence process of TbCSV remain poorly understood. This study analyzed the phylodynamics of TbCSV by examining its nucleotide sequences of the coat protein (CP) gene collected between 2000 and 2022. Using various combinations of priors, Bayes factor comparisons identified heterochronous datasets (3 × 100 million chains) generated from a strict molecular clock and Bayesian skyline tree priors as the most robust. The mean substitution rate of the CP gene was estimated at 6.50 × 10^−4^ substitutions/site/year (95% credibility interval: 4.74 × 10^−4^–8.50 × 10^−4^). TbCSV was inferred to have diverged around 1920 CE (95% credibility interval: 1887–1952), with its most probable origin in South Asia. These findings provide valuable insights for the phylogeography and evolutionary dynamics of TbCSV, and contribute to a broader understanding of begomovirus epidemiology.

## 1. Introduction

Begomoviruses form the largest genus in the family *Geminiviridae*, comprising over 500 members [1,2]. Over recent decades, these viruses have posed significant challenges to agricultural production in tropical and subtropical regions worldwide [3]. Tobacco curly shoot virus (TbCSV), a begomovirus, was first reported in Yunnan, China, in 2002 [4]. Since then, it has been detected across East, Southeast, and South Asia [5]. TbCSV infects several solanaceous crops, including tobacco, tomato, and pepper, as well as various non-cultivated plant species such as *Ageratum conyzoides*, *Manihot esculenta*, *Phaseolus vulgaris*, and *Alternanthera philoxeroides* [3,6,7,8]. In tobacco and tomato, TbCSV infection causes severe leaf curling, leading to significant yield losses.

Like other begomoviruses, TbCSV is transmitted exclusively by the whitefly *Bemisia tabaci* [4]. The TbCSV genome consists of a circular single-stranded DNA (ssDNA) with an intergenic region and six overlapping open reading frames (ORFs) transcribed in opposite directions [9]. The sense strand encodes the V1 and V2 proteins, while the complementary strand encodes the C1, C2, C3, and C4 proteins [10]. V1, the viral coat protein (CP), is responsible for packaging viral particles, and also functions as the viral movement protein, facilitating the transport of the viral DNA/protein complex from the nucleus to the cytoplasm [11]. Some field isolates of TbCSV are associated with a beta-satellite (tobacco curly shoot beta-satellite, TbCSB), which exacerbates disease symptoms in infected plants [12,13].

Although recent research has advanced the understanding of TbCSV pathogenesis, protein functions, and microRNA interactions [14,15,16,17], the origin and evolution of TbCSV remain poorly understood. Here, a Bayesian phylodynamic approach was employed in this study to investigate the phylogeography and evolutionary history of TbCSV, using the CP nucleotide sequences of this virus.

## 2. Materials and Methods

CP gene sequences of 118 TbCSV isolates with known sampling dates and geographic locations were retrieved from GenBank (15 July 2024). These isolates were collected from Bangladesh, China, India, and Myanmar between 2000 and 2022 (Figure 1a). Detailed information about these isolates, including host species and sampling times, is provided in Appendix A. For isolates with multiple sequenced clones, only one randomly selected clone was retained to ensure the accuracy of the results. Following this down-sampling, a final dataset of 95 CP gene sequences was assembled. Multiple sequence alignments were performed using the MUSCLE codon-based algorithm [18] implemented in MEGA X [19]. Excluding the stop codons, the sequence alignment comprised a total of 771 nucleotides.

Containing recombinants can lead to inflated and false-positive evolutionary rates, so the RDP 4.97 package, which includes the RDP, GENECONV, BOOTSCAN, MAXCHI, CHIMAERA, SISCAN, and 3SEQ methods, was used to perform recombination analysis [20]. Only recombination events detected by at least four of the seven algorithms, with an associated *p*-value of less than 10^−6^, were accepted. The complete dataset was used for subsequent dating analyses as no significant recombinants were identified in our dataset.

To mitigate potential biases in molecular dating due to the confounding of temporal and genetic structures [21], we first conducted a Mantel test to evaluate the correlation between pairwise genetic distances and sampling date differences. The *p*-value was calculated based on 1000 permutations, with significant evidence of confounding detected (*p* = 0.012, Appendix A). Following this, we used a clustered-permutation date-randomization test to further assess the temporal signal in the dataset [22].

Additionally, we employed a BETS (Bayesian Evaluation of Temporal Signal) analysis [23] to validate the temporal signal. This method estimates log marginal likelihoods using generalized stepping-stone sampling [24] and compares the fit of two nested models: a heterochronous model (*M*_het_), which incorporates the actual sampling times, and an isochronous model (*M*_iso_), where all samples share the same time points. Bayes factors (BF) were calculated to assess model support [25].

The time of the most recent common ancestor (tMRCA) and the substitution rate of TbCSV were estimated using the Bayesian coalescent approaches implemented in BEAST 1.10.4 [26] under the K2P+Г4+I substitution model, determined by ModelTest-NG [27]. We used three tree priors (constant-size, Bayesian skyline, and exponential growth) and two clock models (strict and relaxed molecular clock) to calculate marginal likelihoods using path sampling and step-stone sampling [28]. The combination of the Bayesian skyline tree prior and the strict clock model was identified as the best fit for our dataset (Appendix A). The sampling time of the TbCSV isolate was used to calibrate the molecular clock. We ran three independent Markov chains, each 100 million iterations, with states sampled every 10,000 generations after discarding the initial 10% of samples as burn-in. To ensure adequate sampling, we used Tracer 1.7 to check the effective sample sizes of all parameters [29].

To trace the origin of TbCSV, a discrete Bayesian phylogeographic model was employed to infer the geographic location of the root state. Due to their geographical proximity, India and Bangladesh were combined into a single discrete state. Consequently, only three geographical regions (China, Myanmar, and India/Bangladesh) were selected and coded as discrete states. The Bayesian MCMC analysis was carried out in BEAST 1.10.4, applying the same substitution models as previously described. Posterior distributions of the parameters were estimated through Markov chain Monte Carlo (MCMC) sampling, as described above. The maximum clade credibility tree obtained from the MCMC analyses was further processed using Python scripts created by Brynildsrud et al. [30] to explore the migration load and directional trends over time.

In addition, to assess the reliability of the inferred most plausible location at the root node, we conducted a comparison with results obtained from 10 replicated datasets, each of which had the location states randomized across sequences, ensuring a robust evaluation against potential biases.

As an extra layer of verification, maximum-likelihood (ML) phylogenetic analysis was performed using IQ-TREE 2.3.6 [31], incorporating the aligned CP gene sequences of TbCSV alongside 442 other begomoviruses (Appendix A). ML analysis was carried out under the GTR+F+Г_4_+I substitution model, as chosen by ModelFinder [32]. To ensure robust statistical support for the inferred tree topology, node reliability was assessed through ultrafast resampling, with 10, 000 replicates applied.

## 3. Results

### 3.1. Temporal Signal in the Dataset of TbCSV

Our data passed the criterion 1 (CR1) of the clustered date-randomization test (Appendix A) and even met the more conservative criterion 2 (CR2) proposed by Duchene et al. [22], suggesting the presence of a temporal structure in our dataset. Our BET analyses further provided evidence against sufficient temporal signal in our dataset. Compared to the *M*_iso_ (−3688.28), the *M*_het_ yielded a higher marginal likelihood (−3645.61), indicating that the latter provided a better fit to our sequence data. This allows us to perform subsequent molecular dating analyses.

### 3.2. Evolutionary History of TbCSV

Bayesian phylogenetic analysis estimated the mean substitution rate of the CP gene of TbCSV to be 6.50 × 10^−4^ substitutions/site/year (95% credibility interval 4.74 × 10^−4^–8.50 × 10^−4^). The time-scaled maximum clade credibility tree revealed that TbCSV isolates segregate into two distinct clades (Figure 1b). Clade I comprises viral isolates from a wide range of sampling locations, while Clade II includes only two isolates, both originating from Myanmar. TbCSV isolates within Clade I can be further divided into two well-resolved subclades. Subclade Ia consists entirely of TbCSV isolates from China, whereas Subclade Ib contains viral isolates from India, China and Bangladesh.

Bayesian phylogenetic analysis dated the crown group of TbCSV to 1920 CE (Common Era, 95% credibility interval 1887–1952). The most recent MRCAs of subclade Ia and Ib were dated to 1998 CE (95% credibility interval 1993–2005 CE) and 1968 CE (95% credibility interval 1954–1981 CE), respectively. The MRCA shared by subclades Ia and Ib was dated to 1961 CE (95% credibility interval 1946–1978 CE).

### 3.3. The Origin and Migration Pattern of TbCSV

Using discrete phylogeography, South Asia (India/Bagladesh) was inferred as the root state, with high posterior probabilities of 0.57 (Figure 1b). This is outside the range of probabilities (0.23–0.44) obtained in our analyses of 10 location-randomized datasets (Appendix A). The phylogeny of the TbCSV isolates analyzed in this study correlates well with their geographical distribution, suggesting very infrequent gene flow among TbCSV populations in South Asia, East Asia, and Southeast Asia. Indeed, our migration analysis indicates a low level of TbCSV migration in any direction (Appendix A).

Although some begomoviruses are widely distributed across distinct subcontinents or even different continents, most are found only locally. This offers an opportunity to substantiate the South Asia origin of TbCSV through conventional phylogenetic analysis. To achieve this, a phylogenetic tree was generated using the CP nucleotide sequences of 443 begomoviruses. Figure 2 illustrates a monophyletic branch of this tree. Of the 11 begomoviruses included in this branch, only two have been detected in East Asia (including TbCSV), while the remaining nine are found exclusively in South Asia. This suggests that TbCSV is more closely related to South Asian begomoviruses than to those in East Asia.

## 4. Discussion

The genus *Begomovirus* is the largest genus of plant viruses, with many species recognized as major pathogens in crop plants. However, the evolutionary history of most begomoviruses remains poorly understood. This study addresses that gap by applying Bayesian phylodynamic analysis to TbCSV, a begomovirus that poses significant threats to agriculture across East, Southeast, and South Asia.

Begomoviruses have relatively smaller genomes compared to most other plant viruses, which has facilitated the deposition of many full-genome sequences of begomoviruses in GenBank, including those of TbCSV. However, begomoviruses are known for their high recombination potential, complicating the use of phylodynamic methods. To address this challenge, only the CP gene sequences of TbCSV were used in this study. Fortunately, recombination analysis confirmed that the CP gene of TbCSV is free from recombination events, making it suitable for reliable evolutionary analysis.

The substitution rate has been investigated for several begomoviruses, such as African cassava mosaic virus (ACMV), East African cassava mosaic virus (EACMV), tomato severe rugose virus (ToSRV), and tomato yellow leaf curl virus (TYLCV), with estimated rates of 1.3 × 10^−4^, 3.83 × 10^−3^, 5.7 × 10^−4^, and 4.6 × 10^−4^ substitutions/site/year, respectively [33,34,35,36]. Additionally, the substitution rates for two other geminiviruses, digitaria streak virus (DSV) and maize streak virus (MSV), have been reported at 1.27 × 10^−4^ and 4.87 × 10^−4^, respectively [37,38]. This study estimated a mean substitution rate for TbCSV of 6.50 × 10^−4^ substitutions/site/year. This suggests that TbCSV evolves much slower than EACMV, faster than ACMV and DSV, and at a rate comparable to ToSRV, TYLCV, and MSV. While further investigation is needed to understand the implications of these differences, it is clear that the evolutionary rate of geminiviruses does not correlate with host range, geographical distribution or phylogenetic relationship.

First described in 2002, TbCSV might initially appear to be a recently emerged virus [4]. However, this study dated the most recent common ancestor of sampled TbCSV isolates to around 1920—about 80 years before its formal documentation (Figure 1b). This suggests that TbCSV may have gone unnoticed for decades, which is surprising given the severe symptoms it causes in several crop plants. However, this can be explained by its ability to infect a wide range of weed species, indicating that TbCSV may have circulated in weeds before spreading to crops. When and how TbCSV jumped to crop plants is unclear. However, the global spread of the B-biotype *B. tabaci* occurred in 1990s. The polyphagous feeding habits of this whitefly biotype are believed to have facilitated the transfer of many begomoviruses from weeds to crops [39].

Although TbCSV is widely distributed geographically, its sequences have not been sampled evenly, making it difficult to pinpoint the virus’s exact origin. To address this challenge, the study grouped available TbCSV isolates into three broadly defined geographical populations: East Asia, Southeast Asia, and South Asia. Based on this approach, South Asia was inferred as the most probable origin of TbCSV (Figure 1b). This finding is somewhat surprising, given the seemingly greater genetic diversity of TbCSV in China compared to other regions. However, the inference is supported by the results of 10 location-randomized analyses (Appendix A). Additionally, it is reinforced by phylogenetic analysis (Figure 2), which shows a closer relationship between TbCSV and a set of begomoviruses documented exclusively in South Asia.

The findings discussed above, along with the topology of the time-scaled maximum clade credibility tree shown in Figure 1b, suggest a scenario for the emergence and evolution of TbCSV. In this scenario, TbCSV may have circulated in South Asia for an extended period, primarily in weed hosts. Around 1920, one lineage of TbCSV was introduced to Southeast Asia, forming the ancestor of the current Clade II TbCSV. By the 1960s, TbCSV was exported to China, leading to the divergence of Clade I into two subclades: Ia in China and Ib in South Asia. Subsequently, a second migration event occurred, introducing Clade Ib TbCSV into China. The phylogenetic relationship of TbCSV correlates well with its geographical distribution, indicating that gene flow between East Asia, Southeast Asia, and South Asia is extremely low, despite occasional emigration events. Furthermore, certain geographical factors may significantly influence the evolution of TbCSV. In contrast, the phylogeny of TbCSV shows little correlation with host species, suggesting that interspecies gene flow is common among TbCSV populations.

## 5. Conclusions

In summary, the results of this study provide valuable insights into the evolutionary history of TbCSV. Findings such as its infrequent gene flow between geographical locations and frequent gene flow between different hosts may contribute to the development of sustainable management strategies against the virus. These results also enhance our understanding of the epidemiology of begomoviruses, a genus of geminiviruses of growing agricultural significance.

## Figures and Tables

**Figure 1 viruses-16-01850-f001:**
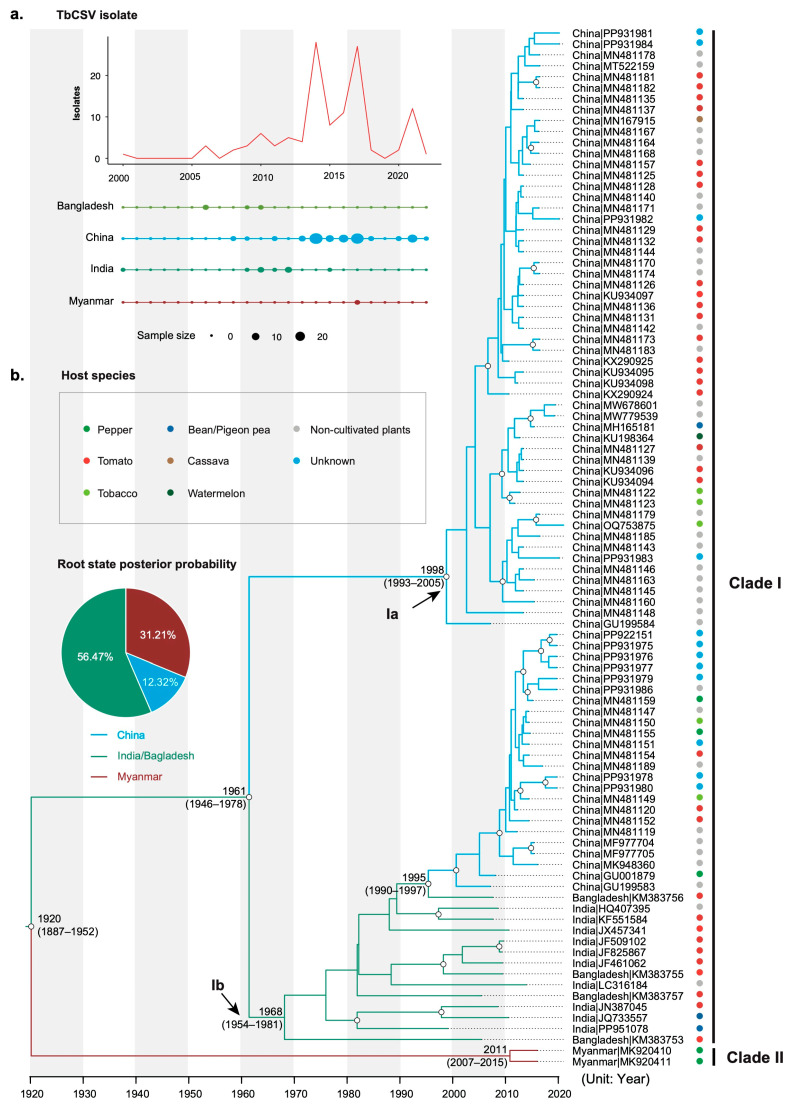
Sampling overview and the time tree of tobacco curly shoot virus (TbCSV). (**a**) Total sample sizes and the distribution of TbCSV isolates across the geographic locations through time (year). Viral isolates from different regions are indicated by a unique color. The dot sizes are proportional to the sample sizes. (**b**) Bayesian maximum clade credibility tree inferred from the CP sequences of TbCSV. The tree topology has been selected to maximize the product of node posterior probabilities and the tree branches have been color-coded according to their geographic origins. Black circles denote strong node support with posterior probability >0.95. The inferred probability of the root for each geographic region is shown in the pie charts.

**Figure 2 viruses-16-01850-f002:**
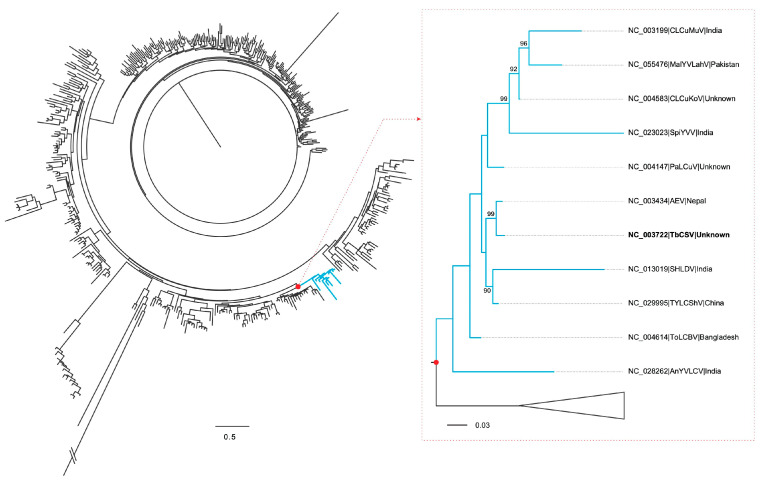
Maximum-likelihood phylogenetic tree of 443 begomoviruses based on the CP nucleotide sequences. Ultrafast bootstrap support values greater than 90% are indicated at key nodes. The scale bar represents the number of nucleotide substitutions per site. Detailed virus names corresponding to the abbreviations in the tree are provided in Appendix A. TbCSV is highlighted in bold font.

## Data Availability

All data used in this study are publicly available on NCBI. A list of the accession numbers used is found in Appendix A.

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
