# Peer review of "Phylogeography and Evolutionary Dynamics of Tobacco Curly Shoot Virus"

_viruses, 2024, doi:10.3390/v16121850_

Round 1
Reviewer 1 Report
Comments and Suggestions for Authors
The manuscript by Long, X. et al. reports the origin and divergence of TbCSV, including the inference of the substitution rate of the coat protein (CP) gene and phylodynamics analysis.
Given the significance of this begomovirus in solanaceous crops across Asia, this work is interesting, and methodologically sound, by using the nucleotide sequences from the CP gene collected between 2000 to 2022, and employing Bayesian phylodynamics and molecular clock methodologies to infer temporal dynamics. Therefore, I think that the study presents a well-structured analysis and shows clear evidence of a temporal structure, which supports the reliability of the molecular clock estimates.
However, despite the approaches used being methodologically solid, the study relies solely on sequencing data retrieved from GenBank. This is a valid and appropriate approach, but it constrains the novelty of the research.
Below are specific suggestions and queries that should be addressed:
Line 46: It is mentioned that this knowledge is "essential for the development of effective and sustainable management strategies."
Please, expand on how the findings specifically contribute to the management and control of TbCSV. For instance, are there any implications for crop protection strategies, breeding resistant varieties, or vector control?
Line 51: Please provide details regarding the genomic fragment of the CP gene analyzed. Size?
It would be valuable to discuss the reliability of using only the CP gene for evolutionary inferences compared to full-length genome analyses. Are there potential biases or limitations in focusing solely on the CP gene?
Lines 176-180: As far as I know, ToRSV is an RNA virus. And possibly, it would be beneficial to consider expanding this discussion including a comparison with RNA plant viruses.
Lines 184-187: The statement "the evolutionary rate of geminiviruses does not correlate with host range, geographical distribution, or phylogenetic relationship" needs further explanation. Please clarify why this is the case. In fact, regarding the divergence analysis and results an intriguing point is about the potential divergence across different host plant species (Fig.1b). is there any chance to perform an analysis of host divergence?
Lines 205-207: At this point, I was wondering if the overrepresentation of sequences from certain geographical regions could introduce bias in your conclusions about the virus's origin in South Asia. It would be helpful to quantify or discuss the extent to which this sampling bias might affect your results.
Author Response
General comments
The manuscript by Long, X. et al. reports the origin and divergence of TbCSV, including the inference of the substitution rate of the coat protein (CP) gene and phylodynamics analysis. Given the significance of this begomovirus in solanaceous crops across Asia, this work is interesting, and methodologically sound, by using the nucleotide sequences from the CP gene collected between 2000 to 2022, and employing Bayesian phylodynamics and molecular clock methodologies to infer temporal dynamics. Therefore, I think that the study presents a well-structured analysis and shows clear evidence of a temporal structure, which supports the reliability of the molecular clock estimates. However, despite the approaches used being methodologically solid, the study relies solely on sequencing data retrieved from GenBank. This is a valid and appropriate approach, but it constrains the novelty of the research. Below are specific suggestions and queries that should be addressed:
RESPONSE: We sincerely thank the reviewer for taking the time to review our manuscript and for providing valuable feedback. We have incorporated the suggested revisions, as detailed in our point-by-point responses below.
Specific comments
Line 46: It is mentioned that this knowledge is "essential for the development of effective and sustainable management strategies." Please, expand on how the findings specifically contribute to the management and control of TbCSV. For instance, are there any implications for crop protection strategies, breeding resistant varieties, or vector control?
RESPONSE: Thank you for highlighting this point. We agree that the specific implications for management strategies were not fully addressed. To avoid uncertainty, we have removed this expression from the manuscript.
Line 51: Please provide details regarding the genomic fragment of the CP gene analyzed. Size? It would be valuable to discuss the reliability of using only the CP gene for evolutionary inferences compared to full-length genome analyses. Are there potential biases or limitations in focusing solely on the CP gene?
RESPONSE: Thank you for the suggestion. We have included a concise description of the CP gene sequence length to provide additional clarity for the phylodynamic analyses. Previous studies, such as those by Awadalla (doi: 10.1038/nrg964) and Sironi et al. (doi: 10.1038/nrg3905), highlight the potential for erroneous estimates of evolutionary rates when recombinants are included in evolutionary analyses. Consequently, the majority of phylodynamic studies to date assume no recombination. However, recombination is known to occur frequently across the complete genome of begomoviruses. To address this issue, our study specifically focused on the recombination-free CP gene for evolutionary analysis. Notably, incorporating recombination remains a persistent challenge with Bayesian phylodynamic frameworks. Despite the relatively short coding sequence of the CP gene in each virus, it has proven to be an invaluable target for evolutionary studies across numerous RNA viruses. For example, Mbewe et al (DOI: 10.1016/j.virusres.2024.199397) demonstrated its utility in analyzing evolutionary dynamics for more than 40 viruses. Similarly, we utilized the CP gene in evolutionary analysis of importance plant RNA viruses, including pepper mild mottle virus (DOI: 10.1016/j.virusres.2018.08.006), potato virus S (DOI: 110.1016/j.virol.2018.09.022), tomato mosaic virus (DOI: 110.1016/j.virol.2020.12.009), rice stripe virus (10.3390/v14112547), papaya ringspot virus (DOI:10.1111/ppa.12942).
Lines 176-180: As far as I know, ToRSV is an RNA virus. And possibly, it would be beneficial to consider expanding this discussion including a comparison with RNA plant viruses.
RESPONSE: We acknowledge an error in the previous version of the manuscript. The virus in question is tomato severe rugose virus (ToSRV), not tomato ringspot virus (ToRSV). This mistake has been corrected in the revised manuscript.
Lines 184-187: The statement "the evolutionary rate of geminiviruses does not correlate with host range, geographical distribution, or phylogenetic relationship" needs further explanation. Please clarify why this is the case. In fact, regarding the divergence analysis and results an intriguing point is about the potential divergence across different host plant species (Fig.1b). Is there any chance to perform an analysis of host divergence?
RESPONSE: We greatly appreciate the suggestion and agree that performing an analysis of host divergence would make the results more compelling. However, as the current study focuses on the phylogeographic history of TbCSV, we plan to address host divergence in a subsequent study.
Lines 205-207: At this point, I was wondering if the overrepresentation of sequences from certain geographical regions could introduce bias in your conclusions about the virus's origin in South Asia. It would be helpful to quantify or discuss the extent to which this sampling bias might affect your results.
RESPONSE: We share the reviewer concerns on sampling bias. Indeed, a region may be identified as the source of the virus due to over-representation in the dataset. In our study, China accounted for the largest sample size, with over 60% of the viral isolates originating from the country. However, the BSSVS analysis was not influenced by sample size, as South Asia was identified as the viral origin instead of China. To further address this issue, we have conducted additional Bayesian phylogenetic analyses using a region-randomized approach to assess the reliability of the inferred root node’ location. Importantly, the probability of South Asia being identified as the root node from the original data set fell outside the range of probabilities obtained from the 10 subsampled data sets. These results are presented in Fig. S2 of the original manuscript.
Reviewer 2 Report
Comments and Suggestions for Authors
Remarks:
Lines 185-187: “It is clear that the evolutionary rate of geminiviruses does not correlate with host range, geographical distribution or phylogenetic relationship”. This sentence is somewhat contradictory to the following: “certain geographical factors may significantly influence the evolution of TbCSV” (lines 217-218).
Lines 187-188: “Notably, this observation mirrors the patterns seen in positive-sense RNA viruses, such as potyviruses”. I would recommend being careful with such a statement, at least in relation to potyviruses, for example potato virus Y or plum pox virus.
Editorial note:
Line 164: “Begomoviruses are the largest genus of plant viruses”. In my opinion, the use of terminology is not entirely correct in this context. It is better to use the singular instead of the plural: “Genus Begomovirus … etc.” or something like “representatives of the genus Begomovirus”.
Author Response
Comment
Lines 185-187: “It is clear that the evolutionary rate of geminiviruses does not correlate with host range, geographical distribution or phylogenetic relationship”. This sentence is somewhat contradictory to the following: “certain geographical factors may significantly influence the evolution of TbCSV” (lines 217-218).
RESPONSE: We appreciate the reviewer’s observation. We do not view the two sentences as contradictory but rather as addressing different aspects of virus evolution. The statement regarding the evolutionary rate refers specifically to the rate at which a virus diverges at the nucleotide level. While the evolutionary rate of geminiviruses does not appear to correlate with host range, geographical distribution, or phylogenetic relationships, geographical factors may still play a role in the broader evolutionary processes, influencing the diversification of TbCSV across different locations. Thus, while the rate of evolution may be similar across regions, geographical factors seem to drive the diversification of TbCSV, which is reflected in its phylogeographic history.
Comment
Lines 187-188: “Notably, this observation mirrors the patterns seen in positive-sense RNA viruses, such as potyviruses”. I would recommend being careful with such a statement, at least in relation to potyviruses, for example potato virus Y or plum pox virus.
RESPONSE: Thank you for highlighting this point. To avoid uncertainty, we have removed this expression from the manuscript.
Editorial note: Line 164: “Begomoviruses are the largest genus of plant viruses”. In my opinion, the use of terminology is not entirely correct in this context. It is better to use the singular instead of the plural: “Genus Begomovirus … etc.” or something like “representatives of the genus Begomovirus”.
RESPONSE: We thank the reviewers pointed this out and have rephrased the sentence as suggested.
Reviewer 3 Report
Comments and Suggestions for Authors
Long et al submitted a manuscript titled "Phylogeography and evolutionary dynamics of tobacco curly shoot virus" for publication in Viruses. Though the work is brief, it is well done and written. Though it comes within the scope of the journal, the manuscript is not suitable for the impact this journal has created in pushing the field significantly forward. This ms is best suited in subject journals accepting brief manuscripts.
Author Response
Comment:
Long et al submitted a manuscript titled "Phylogeography and evolutionary dynamics of tobacco curly shoot virus" for publication in Viruses. Though the work is brief, it is well done and written. Though it comes within the scope of the journal, the manuscript is not suitable for the impact this journal has created in pushing the field significantly forward. This ms is best suited in subject journals accepting brief manuscripts.
RESPONSE: We appreciate the reviewer’s thoughtful review of our manuscript and the recognition of the quality and relevance of our work. Regarding the concern about the suitability of our manuscript for Viruses, we respectfully highlight that similar phylogeographic studies have been published in the journal. For example, our previous work on the origin and dissemination of rice stripe virus was featured in Viruses (DOI: 10.3390/v14112547), emphasizing the journal's interest in research on viral evolution. Additionally, the study by Dr. Jianguo Shen on the spatial diffusion of cherry leaf roll virus (DOI: 10.3390/v14102179) provides a precedent for such studies being within the journal's scope and impactful readership. We believe it aligns well with Viruses’ mission of publishing high-quality research that advances understanding in virology. Furthermore, we would like to highlight that Viruses accepts brief manuscripts, and our submission conforms to this format.
Round 2
Reviewer 1 Report
Comments and Suggestions for Authors
In my opinion, the authors cope adequately with my comments on the previous version. Thus, I'm pleased to see the improvements in combination with complementary revisions. I have no further comments from the particular research perspective.